# Global transcriptomic profiling and mutant analysis suggest linked 3-chlorobenzoate metabolism and activation of an integrative and conjugative element in *Pseudomonas putida*

Roxane Bertholet,[1] Valentina Benigno,[1] Hanna Budny,[1] Anthony Convers,[1] Vladimir Sentchilo,[1] Andrea Daveri,[1] Nicolas Carraro,[1] Jan Roelof van der Meer[1]

**ABSTRACT** Conjugation, the exchange of genetic material between different bacterial species, is an important process underlying adaptation. The reason for this is that conjugative elements carry additional genes potentially providing adaptive benefit to the host, such as antibiotic resistance or pathways for degradation of environmental contaminants. However, there is increasing evidence that the conjugative process itself is also responsive to cellular or environmental cues and may, thus, be under rate-dependent selection. Here, we investigate the integrative and conjugative element ICE*clc* as a model for both metabolic pathway "engrafting" (i.e., how a *Pseudomonas putida*-encoded benzoate pathway is expanded to 3-chlorobenzoate; 3-CBA) and understanding how metabolic cues or states may lead to ICE activation. We compared differences in gene expression of *P. putida* with or without ICE*clc* grown on 3-CBA, benzoate, and succinate, both in exponential and stationary phase conditions. Mutants were constructed in the 3-CBA metabolic pathway, and effects on ICE-transfer were examined. We further deleted parts of a seemingly redundant methyltetrahydrofolate (mTHF) biosynthesis operon that was highly expressed during 3-CBA growth and studied its effects on ICE-activation. Our results demonstrate that 3-CBA metabolism is necessary for ICE activation and reveal an unexpected connection between mTHF pathway induction and ICE*clc* activation and transfer. These findings show how metabolic states may arise that are perceived by the ICE and lead to an increase in its transfer frequency, thereby mobilizing the very genes that partly caused this state to appear.

**IMPORTANCE** Bacterial conjugation is widely appreciated for its diversity, its molecular details, and evolutionary consequences. However, the regulation of the onset of the conjugative process in or by the host cell is frequently taken for granted, whereas any influence of environmental or cellular cues on the rates of conjugation can have drastic consequences on both positive and negative outcomes of conjugation-dependent adaptation (e.g., the distribution of genes for antibiotic resistance). We study here a particular case of an integrative and conjugative element (ICEclc), representative of a widely distributed family of elements pervasive in pseudomonads, which has a modulable conjugation rate in donor populations that is dependent on the proportion of transfer-competent cells arising in stationary phase. Transfer-competent cell appearance permits to quantify and understand conditions favoring ICE activation and transfer, and we show here how the ICE reacts to differences in the physiological states of the host cell under influence of its metabolism of aromatic compounds.

**KEYWORDS** RNA-seq, ICE*clc*, oxidative stress, conjugative transfer, aromatic compound metabolic pathways

**Peer Reviewers** Joao Botelho, Max-Planck-Institut fur Evolutionsbiologie, Plön, Germany; Nathan Fraikin, Université Lyon 1, CNRS, Lyon, France

Address correspondence to Jan Roelof van der Meer, janroelof.vandermeer@unil.ch.

Roxane Bertholet and Valentina Benigno contributed equally to this article. Author order was based on chronological order of involvement in the study.

The authors declare no conflict of interest.

See the funding table on p. 18.

Conjugation allows bacteria to rapidly share and acquire genetic information encoding potential adaptive functions, such as antibiotic resistance or catabolic pathways (1–3). The process is mediated by mobile DNA elements that, in addition to transferring their own DNA, carry auxiliary genes contained within their boundaries (4). One such element, ICE*clc* from *Pseudomonas knackmussii* B13, confers its host with (among others) the genes for chlorocatechol (*clc*) and 2-aminophenol (*amn*) degradation (Fig. 1A) (5). In strains with native benzoate metabolism, incorporation of the *clc* genes expands their ability to use 3-chlorobenzoate (3-CBA) as sole carbon and energy source (6–10). This process of metabolic complementation works because native benzoate dioxygenase has a sufficiently broad specificity to convert 3-CBA into the corresponding 3- and 4-chlorocatechols (11). Chlorocatechols are not transformed by the (native) catechol 1,2-dioxygenase pathway, which catalyzes the transformation of catechol arising as intermediate from benzoate (12). If strains express the *clc* genes, however, they can productively grow on 3-CBA or other chloro-aromatic compounds (Fig. 1B) (13).

By acquiring new catabolic genes via lateral gene transfer, bacteria can, thus, potentially enhance the range of metabolizable compounds. Proper expression of the acquired pathways is not guaranteed and may lead to metabolic side products or cross-induction of competing pathways in the new host. This pathway implementation is a major well-known problem in bioengineering for the design and construction of new metabolic pathways but may also be expected in cases of naturally transferred genes such as the *clc* and *amn* genes of ICE*clc*. For example, the new proteins may not be produced in the optimal stoichiometry by the cell (14), or at the appropriate timing, since they did not co-evolve with the networks and pathways of the new host (15, 16). The expression of native pathways may interfere with those of the acquired metabolic genes through cross-activation by non-selective regulatory proteins (5). Furthermore, their expression may have disruptive effects and be costly for the cell (17), lead to temporary accumulation of toxic intermediates (11), or demand excess counterbalancing cofactors (15).

ICE*clc* is not only an interesting system to examine pathway "engrafting" as an extension from benzoate to 3-CBA metabolism in a new host, but, curiously, growth on 3-CBA is also the only known trigger for ICE*clc* transfer itself. ICE-transfer genes are activated in a small subpopulation of transfer-competent (tc) cells in stationary phase populations as a consequence of induction of a bistable switch (18, 19). ICE excision, however, only takes place once tc cells acquire new nutrients and restart cell division (20). tc Cells with excised ICE can replicate the ICE-circular DNA, and individual ICE copies can transfer independently from a single donor cell (21). Growth on 3-CBA results in the highest spontaneously achieved subpopulation with activated cells (3%–5%, with less than 0.1% on benzoate) (22), suggesting there may be some link between 3-CBA metabolism and ICE transfer activation. To better understand these links, we examined host pathways and factors that are specifically expressed during or after growth on 3-CBA, independently of ICE*clc* and compared them with those expressed during growth on substrates that are not known to trigger ICE activation (succinate and benzoate). We specifically compared benzoate and 3-CBA since both are transformed by the same dioxygenase and dihydrodiol dehydrogenase to catechol and chlorocatechol, respectively (12). To untangle any role of ICE*clc* in this, we compared isogenic strains of *Pseudomonas putida* having either a single integrated copy of the ICE*clc* element or a single integrated copy of only the *clc* gene cluster from the ICE, allowing it to grow on 3-CBA (23). We further produced deletions of the *ben* and *clc* pathways genes on the *P. putida* chromosome and in the integrated ICE, respectively. Noticing strong induction of a putative *P. putida* operon for methyltetrahydrofolate (mTHF) metabolism under growth on 3-CBA, we also made seamless deletions in parts covering this operon. Genome-wide gene expression differences in the various strain combinations were analyzed and compared by sequencing of reverse-transcribed (ribosomal-RNA depleted) total mRNA (RNA-seq) of cultures growing on 3-CBA, succinate, and benzoate individually, or on

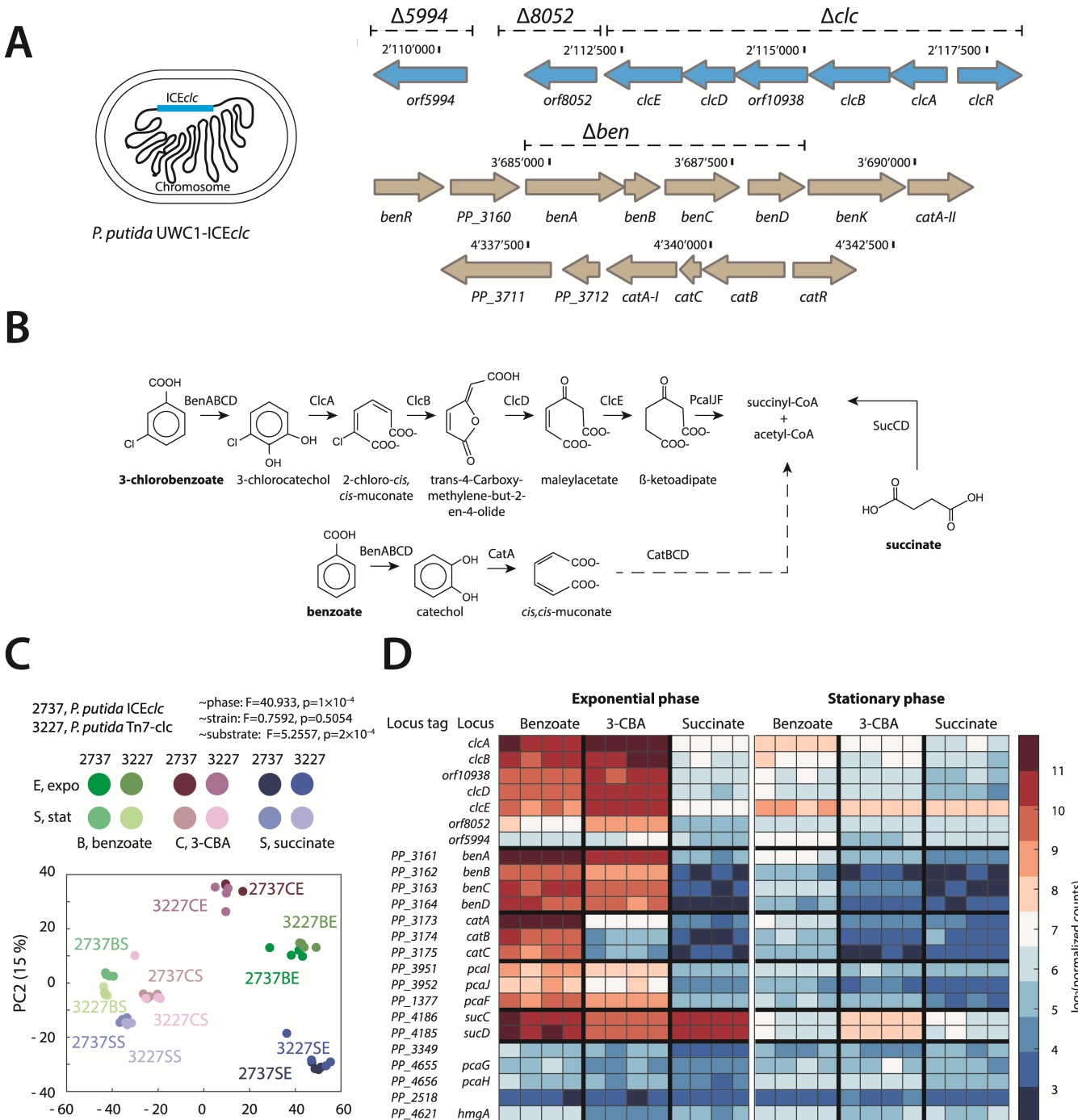

**FIG 1** Relevant pathways of aromatic compound metabolism in *P. putida* with or without integrated ICE*clc*. (A) Genetic organization of the *ben-*, *cat-*, and *clc*-genes in *P. putida* UWC1 carrying one integrated copy of ICE*clc*. The deletions produced in this study are indicated with dashed lines. (B) Overview of the degradation steps of benzoate, 3-CBA, and succinate with the enzymes involved. (C) Principal component analysis of pseudo reference-normalized expression levels in *P. putida* ICE*clc* and *P. putida* without the ICE but with a single copy Tn*7*-insertion of the *clc*-genes. Percentages indicate the proportion of the variance explained by the principal component. Test: *adonis* permutation (*n* = 10,000). (D) Gene expression in *P. putida* ICE*clc* (as log$_2$ normalized counts) for the enzymes in (B) measured by RNAseq. Each row corresponds to a gene, and each column is a replicate (four replicates per condition).

mixtures of succinate and 3-CBA, both in exponential and stationary phase (when ICE activation takes place [24]). This was complemented by analysis of ICE*clc* transfer rates and quantification of tc-cell proportions for the various donors and conditions. Our

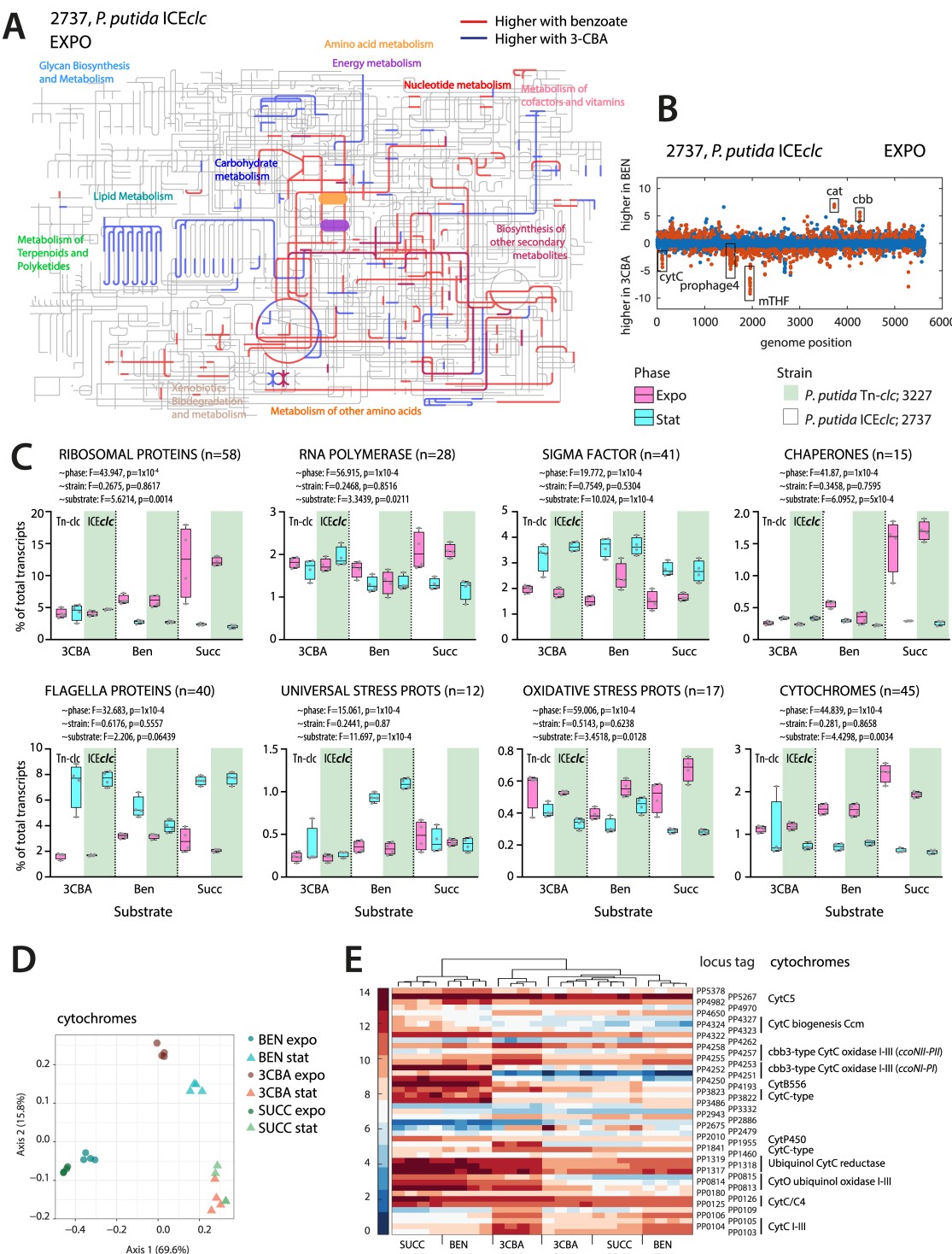

**FIG 2** Specific transcriptome signatures of *P. putida* on different carbon sources and growth phases. (A) Interactive Pathways Explorer (iPath) map for visualization of higher expressed pathways on benzoate (red) and higher expressed pathways on 3-CBA (blue). (B) Scatter plot of the log$_2$ fold-change for the comparison of *P. putida* ICE*clc* (strain 2737) on benzoate vs 3-CBA in exponential phase. Each dot represents a gene along the genome position. Red dots are genes that are statistically significantly differentially expressed (adjusted *P*-value < 0.05; fold change > 2). (C) Grouped transcript levels from ribosomal proteins (*n* = 58), RNA polymerase (*n* = 28), sigma factor (*n* = 41), chaperones (*n* = 15), flagella (*n* = 40), universal stress (*n* = 12), oxidative stress (*n* = 17), and cytochrome proteins (*n* = 45) as percentage of the total transcript levels for both strains, substrates, and growth phases (symbols, individual replicates). Test: *adonis* permutation (*n* = 10,000). Gene assignment to groups listed in Table S5. (D) Principal component analysis of cytochrome genes expression levels. Percentages indicate the proportion of the variance explained by the principal component. (E) Clustergram of cytochrome gene expression (as log$_2$ normalized counts).

results clearly implicate 3-CBA metabolism in ICE-activation and further uncover an unexpected link of mTHF pathway expression and ICE*clc* transfer induction.

## RESULTS

### Global transcriptomic differences in the presence or absence of ICE*clc*

In order to disentangle differences in host gene expression in response to substrate or as a consequence of having an integrated ICE*clc*, we first compared global transcriptomes of two *P. putida* strains, one with a single integrated ICE*clc* copy (*P. putida* ICE*clc*, strain 2737, Table S1) and one with only a single integrated copy of the *clcRABCD* operon (isolated from the ICE and reintroduced on a mini-transposable element; *P. putida* Tn7-*clc*; strain 3227, Table S1), which enables growth on 3-CBA. Cultures were grown on 10 mM succinate, 1 mM 3-CBA, or 1 mM benzoate, and total RNA for RNA-seq was isolated in exponential and stationary phases. Pseudo-reference normalized read counts across all the samples showed low replicate variability and clear consistent global transcriptome clustering across growth conditions (Fig. 1C; *adonis* permutation test [10,000], $F = 40.933$, $P = 1 \times 10^{-4}$ for stationary vs exponential phase). Samples further clustered according to carbon source ($F = 5.2557$, $P = 2 \times 10^{-4}$), but there was no significant effect of the host strain (*P. putida* ICE*clc* or *P. putida* Tn7-*clc*; $F = 0.7592$, $P = 0.5054$). Among all the exponential-stationary phase comparisons, 400 genes (7.6% of the total number of genes) were commonly differentially expressed (DE) with an average of 16-fold change difference of their expression level. Among those genes, 210 were consistently more and 165 were lower expressed in exponential than in stationary phase (Table S2). Across all comparisons, both host strains (with or without ICE*clc*) only differed in 35 expressed host genes, apart from the ICE-genes itself (Table S3). This showed that the presence of the ICE*clc* element itself had a limited impact on the expression of host genes under three carbon substrates and two growth phases. The close similarity of both strains on the three substrates was further evident from population growth kinetics (Fig. S1).

The expression of the respective pathway genes in *P. putida* ICE*clc* followed the main imposed substrate conditions but also showed a number of crosswise pathway interferences (Fig. 1D). As expected (Fig. 1A and B), cells growing on benzoate showed strong expression of the *ben-*, *cat-*, and *sucCD*-genes, but they also induced the *pca*-genes and the ICE-located *clc*-operon. In contrast, cells growing on 3-CBA also expressed the *ben-* and the *sucCD*-genes and the *clc*-operon, but the *cat*-genes were poorly induced (Fig. 1D). Of all these, only the *sucCD*-genes were expressed during growth with succinate. In stationary phase cells, metabolic pathway expression decreased by $2^4$-$2^5$-fold (Fig. 1D), indicating that the primary substrates were depleted.

To further describe differences in gene expression of cells growing on benzoate or 3-CBA, we extended specific pathway induction to a global metabolic overview. A total of 705 genes in exponential phase was higher expressed on 3-CBA and 647 higher on benzoate (Table S4). KEGG-database orthology (KO)-annotated genes higher expressed on benzoate mapped to central amino acid and energy metabolism (Fig. 2A, red reactions), whereas those on 3-CBA affected several but less well-defined branches (Fig. 2A, blue lines). This was further evident from StringDB-analysis that additionally showed significant enrichment on benzoate of terms related to ABC transporters, quorum sensing, and chemotaxis (Table 1), which altogether may reflect the overall faster growth rate on benzoate (Fig. S1). In contrast, no specific KEGG term was significantly enriched among the higher expressed genes on 3-CBA, but a number of identifiable gene clusters appeared from a global comparative genome position plot (Fig. 2B). Notably, this consisted of a large contiguous gene cluster (PP_1943-PP_1957) that contains gene orthologs for methyltetrahydrofolate synthesis (Fig. 2B, mTHF) and a region encompassed by the *P. putida* prophage 4 (Pspu28; PP_1532-PP_1584). Induction of this prophage has been observed before (23), but we show here that this is not taking place on succinate or benzoate, and is independent of the presence of the ICE*clc* (Table S4).

**TABLE 1** KEGG term enrichment during growth on benzoate vs 3-CBA[a]

| Term ID | Term description | Observed gene count | Background gene count | False discovery rate |
|---|---|---|---|---|
| ppu02010 | ABC transporters | 48 | 190 | 0.00096 |
| ppu02030 | Bacterial chemotaxis | 17 | 46 | 0.0096 |
| ppu02024 | Quorum sensing | 24 | 79 | 0.0096 |
| ppu01120 | Microbial metabolism in diverse environments | 58 | 283 | 0.0096 |

[a]StringDB-KEGG enrichment using list of significantly upregulated genes (Log$_2$-fold change > 2 and adjusted $P$-value < 0.05) on benzoate vs 3-CBA.

Grouped by identifiable protein classes, exponential growth on succinate entailed the highest proportions of transcripts for ribosomal proteins (~12% of all transcripts, $n = 58$; Fig. 2C), followed by benzoate (6%) and 3-CBA (4%, Fig. 2C). As expected, the proportion of ribosomal protein transcripts largely decreased in stationary phase (2%–3%), but only for succinate and benzoate cultures. Cultures on 3-CBA retained significantly higher proportions of ribosomal protein transcripts (5%) in stationary phase, which was not an effect of bearing or not the ICE (Fig. 2C). Also, grouped transcripts for RNA polymerase, sigma factors, chaperones, universal stress proteins, flagella, cytochromes, and oxidative stress proteins (groups listed in Table S5) showed significant substrate and growth phase effects, but no effect of host strain (Fig. 2C). Grouped sigma factor expression was higher in stationary phase (Fig. 2C). Summed chaperone transcripts were highest during exponential growth in succinate and benzoate and lower in stationary phase, but for 3-CBA, this was reverse (Fig. 2C). In contrast, transcripts of universal stress proteins were specifically induced in stationary phase with benzoate as substrate, but not with 3-CBA or succinate (Fig. 2C). Transcripts of oxidative stress proteins (e.g., peroxidases, catalase, superoxide dismutase) were all highest in exponentially growing cells (Fig. 2C), irrespective of substrate. Finally, transcripts from cytochromes, representative for respiratory activity, were highest in exponential phase of succinate-, followed by benzoate- and 3-CBA-grown cells (Fig. 2C).

The different expression of gene clusters for cytochromes (e.g., *cbb* and *cytC*) was clearly visible in a global comparison of benzoate- vs 3-CBA-growing cells (Fig. 2B), and the expression of genes under a collective locus description of *cytochrome* as keyword among replicates separated well between exponential and stationary phase cultures, as well as substrate (Fig. 2D). Seen in detail, the expression of genes annotated with *cytochrome* was quite different in exponentially growing cells on succinate and benzoate, or 3-CBA, with much lower expression of *cytC* and *cytB556* (PP_3486, PP_3822, PP_3823, PP_4193, Fig. 2B and E), *ccoNI-PI* cbb3-type CytC oxidase (PP_4250-PP_4253), and *cytO* ubiquinol oxidase I-III (PP_0813-PP_0815). In contrast, seven genes (PP_0103-PP_0109) encoding an alternative cytochrome *c* oxidase and the genes for the alternative cbb3-type CytC oxidase (PP_4255-PP_4258) were higher expressed during 3-CBA growth (Fig. 2E).

Overall, our data showed that despite their molecular similarity, 3-CBA and benzoate elicited different responses in growing and stationary phase cells, which may partly be the effect of a slower growth rate on 3-CBA in general or differences in their metabolism and metabolic support pathways. The presence of ICE*clc* in the host was not responsible for the observed expression differences, indicating that the ICE*clc* is more likely to react to host expression differences rather than influencing the host (which could have confounded our interpretation of 3-CBA induced activation of the ICE in stationary phase).

## Impact of 3-CBA metabolism on ICE*clc* activation

In order to investigate the interplay between 3-CBA metabolism, global gene expression, and ICE*clc* activation, we constructed a variety of mutant *P. putida* with interruptions in the *clc*- or *ben*-genes (Fig. 1A; Table S1). Genes in the ICE*clc* core region were highly expressed in stationary phase after growth of *P. putida* with wild-type ICE on 3-CBA, but not benzoate or succinate (Fig. S2). As expected, strains without the *clc*- or the *ben*-genes were unable to grow on 3-CBA as sole carbon and energy source, and only

the full complementation of the Δ*clc* strain with a mini-Tn*7* containing *clcRABDE* was again able to do so, albeit with a longer lag phase (Fig. 3A). To compare the strains for effects of (partial) 3-CBA metabolism by transcriptomics, we cultured them on minimal media supplemented with 10 mM succinate or a mix of 10 mM succinate and 3 mM 3-CBA (mixed carbon source, Fig. 3A). Under mixed carbon source conditions, both wild-type (7260) and mini-Tn7-*clcRABDE* complemented (7301) strains showed diauxic growth (Fig. 3A), which we assumed corresponded to initial succinate utilization and the subsequent later growth phase emerging as cells begin metabolizing 3-CBA. The onset of 3-CBA metabolism in the presence of succinate for the wild type started after ca. 25 h in stationary phase, whereas the complemented strain 7301 had an additional ca. 30 h delay (Fig. 3A). Neither the *clc*-deletion mutant (7262) nor the *ben*-deletion mutant (7434) showed this second growth phase although the *clc*-mutant produced a dark brown color as a result of accumulating chlorocatechol (Fig. 3A). Irrespective of growth substrates, stationary phase transcriptome samples clustered separately from those in exponential phase (Fig. 3B; *adonis* permutation test [10,000] $F = 88.1704$, $P = 1.0 \times 10^{-4}$). The genes mostly responsible for the separation of exponential and stationary phases were primarily associated with active cellular growth, including ribosomal proteins, ATP synthases, and elongation factors (Table S6). The second separation in principal components was dependent on the growth substrate (Fig. 3B), which is largely driven by differential expression of numerous transcriptional regulators (Table S6). Consistent with the previous transcriptome results, the *sucCD* genes exhibited high expression levels on succinate, but also on mixed substrate exponential growth conditions (Fig. 3C). Their expression remained elevated during the stationary phase on mixed substrate, consistent with the continued activity or metabolism of 3-CBA leading to succinate as intermediate (Fig. 1B). As expected, the deletion of the *ben* genes abolished *clc*-gene expression (its induction being dependent on the formation of 2-chloro-*cis,cis*-muconate, Fig. 1B [25]) and the *clc*-complemented strain only partly restored *clc*-expression (possibly due to the longer secondary lag phase for 3-CBA metabolism under the mixed substrate conditions, Fig. 3A and C). The expression of the *ben*-genes was also affected in the absence of *clc*-genes, which may be due to cross-regulation by ClcR on the *ben*-promoter (26). Further cross-regulation may occur at the *pcaIJF* genes and *catA-II* (Fig. 3C), the latter located downstream of the *ben* operon and encoding an additional catechol-1,2-dioxygenase, which was shown to help keep intracellular concentrations of catechol low to prevent its toxicity (27, 28).

To compare ICE-transfer from donors impaired in 3-CBA metabolism, we introduced a constitutively expressed Km-resistance cassette on the ICE, enabling selection for Km-resistance in transconjugants (instead of selecting for growth on 3-CBA). Growth of *P. putida* wild type on the mixed substrate condition was as efficient as growth on 3-CBA alone in producing donors that can conjugate ICE*clc* to ICE-free *P. putida* recipient, whereas no measurable transfer was detected when donor cells were grown on succinate alone (Fig. 3D). This correlates to higher expression of ICE*clc* core genes (Fig. S2) and formation of the tc state (24). Compared to the wild type, the deletion of the *ben* genes reduced ICE*clc* transfer rates by a factor of 10, but in the absence of the *clc* genes, this decreased further by an average of almost 50-fold (Fig. 3E). All mutant strains transferred at statistically significantly lower rates than the wild type (Fig. 3E, one-way ANOVA, $P < 0.0001$). Transfer was not restored to wild-type levels in the *clc*-complemented strain, which may be due to its slower onset of 3-CBA metabolism than the wild type itself when grown on the mixture of succinate and 3-CBA (Fig. 3A). Proportions of tc-cells in stationary phase cultures on mixed substrate conditions were quantified by fluorescence microscopy using a reporter gene (*echerry*) placed under the control of the ICE promoter $P_{inR}$ (24). These proportions were significantly lower in the *clc*- and *ben*-deletion mutants, but not in strains carrying deletions of the genes downstream of *clc* on the ICE (Δ*orf5994* and Δ*orf8052*, Fig. 1A and 3F). Collectively, this indicated that metabolism of 3-CBA is needed to produce the conditions under which ICE*clc* tc-cells develop in stationary phase.

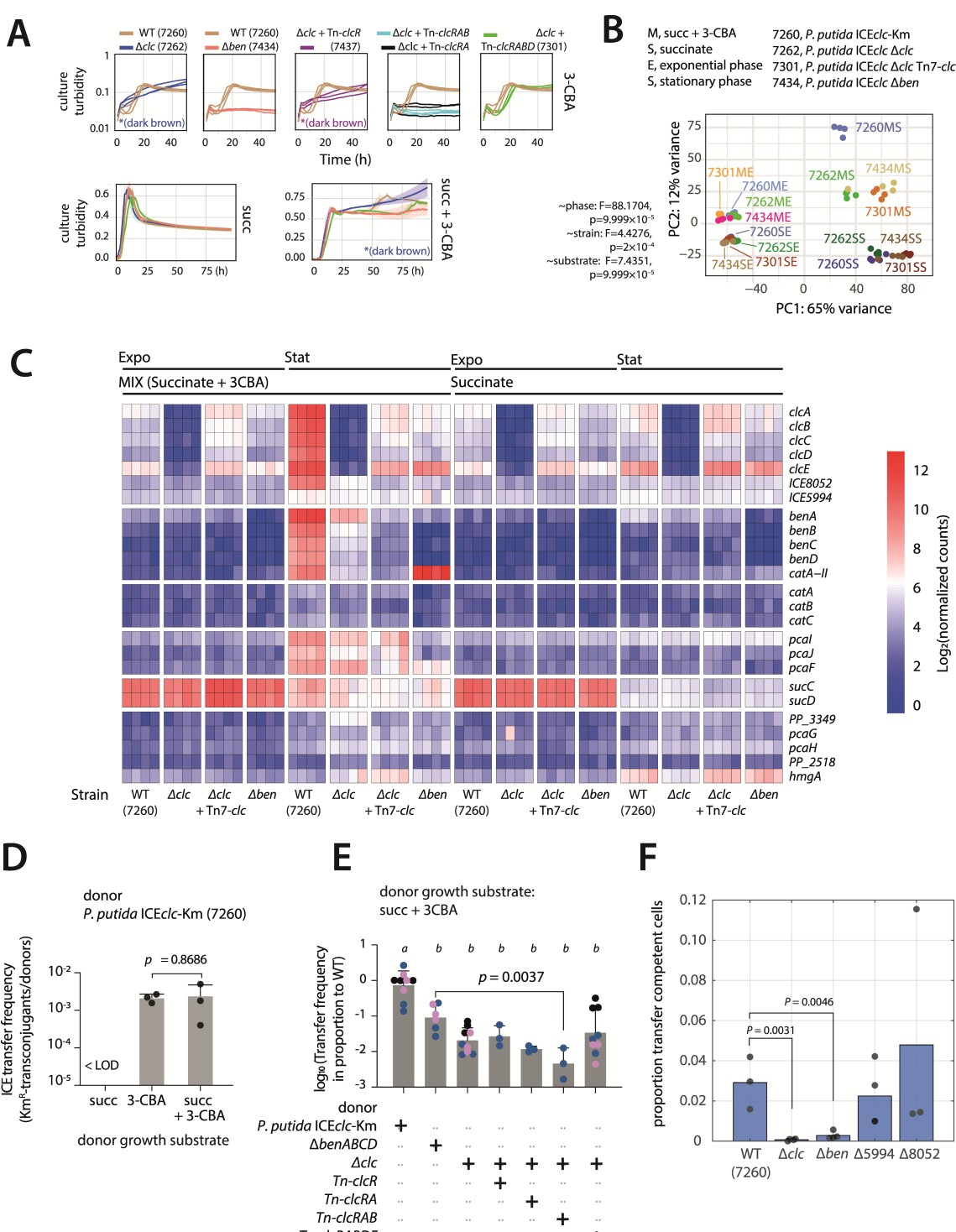

FIG 3  Impact of 3-CBA metabolism on ICE*clc* activation and transfer. (A) Growth of *P. putida* ICE*clc*-Km prototroph for 3CBA metabolism (strain 7260) and derivative mutants impaired in 3-CBA metabolism on 3-CBA, succinate, and succinate + 3 CBA. *(dark brown) indicates the absence of observed cell growth but increase of culture absorbance due to accumulating chlorocatechols and their oxidation to a dark brown colored product. (B) Principal component analysis of pseudo reference-normalized expression levels. Test: *adonis* permutation (*n* = 10,000). (C) Gene expression (as log$_2$ normalized counts) for the enzymes involved in 3-CBA, benzoate, and succinate metabolism measured by RNAseq. Each row corresponds to a gene, and each column is a replicate (four replicates per condition). (D) Transfer frequency for *P. putida* ICE*clc*-Km (7260) donor after growth on 10 mM succinate, 3 mM 3-CBA, or a combination of both carbon sources. *P*-value from two-tailed unpaired *t*-test on the raw data. (E) ICE transfer frequency in proportion to the *P. putida* wild-type ICE*clc*-Km (7260), for the Δ*benABCD* deletion (strain 7434), the Δ*clc* operon (7262), Δ*clc* with a *clcR* complementation (7437), Δ*clc* with a *clcRA* complementation (7439), a *clcRAB* complementation

**Fig 3 (Continued)**

(7438), or a whole *clcRABDE* operon complementation (7301), grown on a mix of 10 mM succinate and 3 mM 3-CBA. A one-way ANOVA with Tukey's multiple comparison tests was performed on $\log_{10}$ transformed values. Groups indicated by b are statistically different in comparison to the wild type indicated by a. Bars in (D) and (E) represent the mean, and error bars represent the standard deviation of the replicates. Individual replicate means are indicated by dots. Dots with the same colors stem from replicates performed on the same day. < LOD, lower than the transfer detection limit ($10^{-7}$). (F) Proportion of ICE*clc* transfer competent cells calculated from microscopy-derived data using QQ-plotting as in reference 22 in *P. putida* wild-type ICE*clc*-Km (7260), the Δ*benABCD* deletion (strain 7434), the Δ*clc* operon (7262), the Δ*orf5994-5512* deletion (strain 7525), and the Δ*orf8052* (7527); all carrying the tc-fluorescent P$_{inR}$-*echerry* reporter and grown on 10 mM succinate and 3 mM 3-CBA to late stationary phase. *P*-values from two-sample one-sided *t*-test.

## Disruption of 3-CBA catabolism alters host global expression

Mutations in the 3-CBA metabolic pathway genes produced several clear effects on global gene expression outside its own metabolism. Summed transcripts for *oxidative stress* genes were particularly elevated in wild type and Δ*clc*-mutants under mixed substrate conditions in stationary phase, likely reflecting the accumulation of chlorocatechols in the culture medium (Fig. 4A and B). In the same strains and conditions, summed transcripts for *sigma factors* were lower expressed (Fig. 4B). In contrast, summed transcripts for *universal stress proteins* were not differentially abundant in dependence of mutant strain or presence of 3-CBA in the growth medium (Fig. 4A).

In pairwise comparisons with the wild type, 12 genes were significantly higher expressed in the Δ*clc*-mutant in stationary phase under mixed substrate conditions (most lacking functional annotation; Fig. 4C; Table S7), whereas genes on ICE*clc* were significantly lower expressed, as expected. ICE*clc* core genes were also uniformly lower expressed in the Δ*ben*-mutant (Fig. 4C), in agreement with the absence of tc cells (Fig. 3F) and decreased ICE transfer (Fig. 3E). In contrast, the complemented Δ*clc*-mutant (strain 7301) showed primarily lower expression of the alternative mTHF pathway genes (Fig. 4C). Across all mutant backgrounds, the gene PP_3089 was consistently lower expressed compared to the wild type. AlphaFold structure prediction and comparison identified PP_3089 as a member of the Hcp1 family of Type VI Secretion System effectors.

None of the mutants clearly reproduced the same expression patterns of the *cytochrome* genes in wild-type stationary phase under mixed substrate conditions, such as the elevated expression of *cytCI-III, cytC/C4,* ubiquinol *cytC* reductase, or *ccoNII-PII* (Fig. 4D). This may reflect the inability to metabolize 3-CBA (Δ*ben*), or only partially (Δ*clc*), or with a delay (the complemented Δ*clc* mutant strain 7301).

## An alternate pathway for methyltetrahydrofolate influences ICE expression with 3-CBA

The high expression of a large gene cluster (PP_1943-PP_1957) in the presence of 3-CBA as opposed to benzoate (Fig. 2B) and in comparison of wild type to the Δ*clc*-complemented strain on mixed substrate conditions was unexpected (Fig. 4C). In comparison to succinate alone, this gene cluster was strongly induced on mixed substrate conditions in exponential phase, even though 3-CBA is not yet serving as a carbon-energy source at that time (Fig. 3A), and independent of the completeness of the 3-CBA pathway genes (Fig. 5A). Differences among strains were clearer in stationary phase on succinate and 3-CBA, with the wild type showing even stronger induction across the complete cluster than in exponential phase (possibly because of starting 3-CBA metabolism here) and mutants lagging behind (Fig. 5A). This resembled the induction observed in 3-CBA-, but not on benzoate- or succinate-exponentially growing cultures of *P. putida* wild type carrying ICE*clc* (strain 2737) and without ICE, but with mini-Tn7-*clcRABDE* insertion (strain 3227; Fig. 5B). Comparison to that experiment also confirmed that a "true" stationary phase culture even after growth on 3-CBA only keeps minor induction of the mTHF pathway genes (Fig. 5B), whereas a late sampling in the diauxic phase of the strain grown on the mixture of succinate + 3 CBA must still be in equivalent of 'exponential' (or possibly early stationary) phase (Fig. 5A, labeled "STAT").

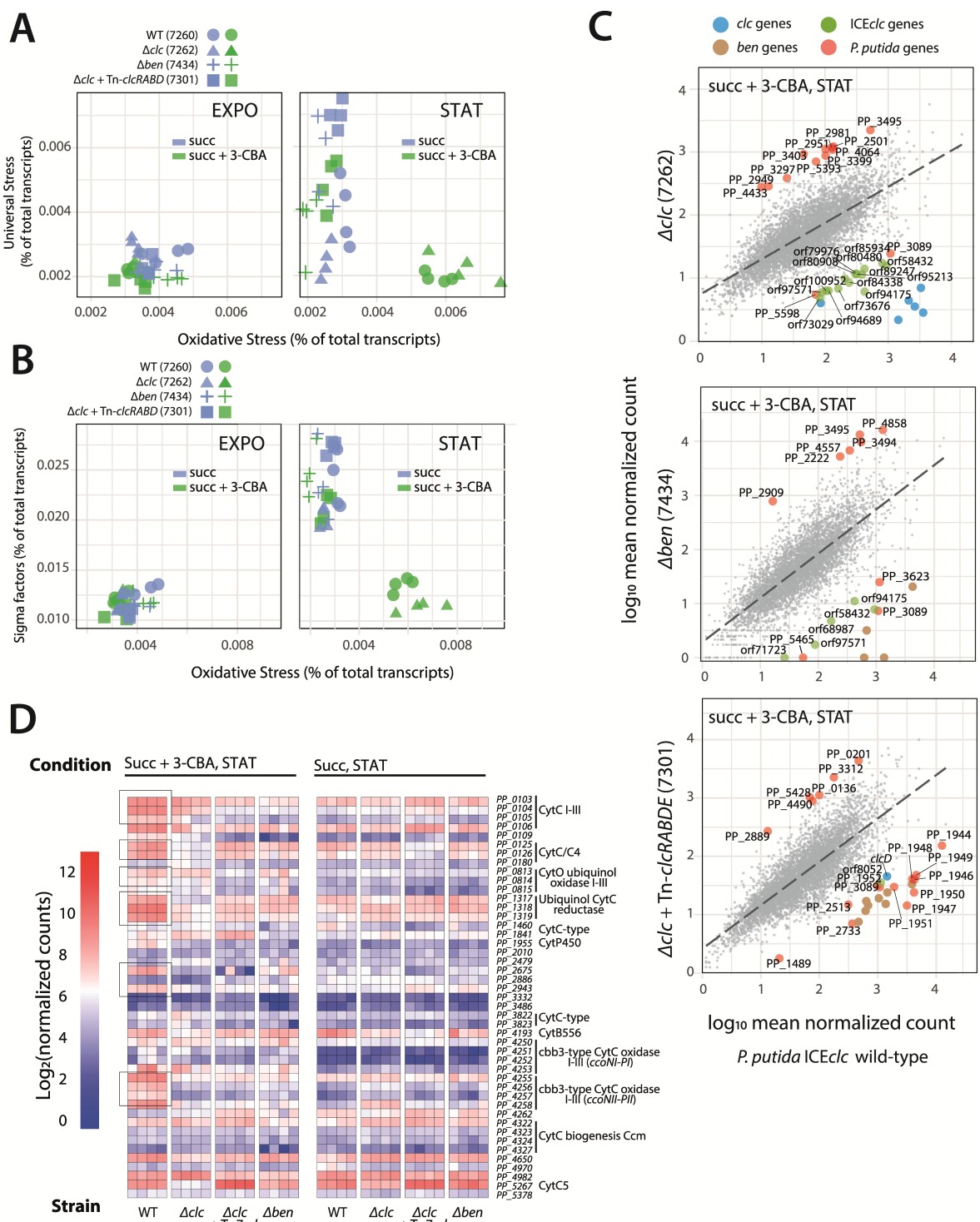

**FIG 4** Influence of 3-CBA metabolism interruption on *P. putida* transcriptional profiles. (A and B) Grouped transcript levels from sigma factors (*n* = 41), universal stress (*n* = 12), and oxidative stress proteins (*n* = 17) as percentage of the total transcript levels for each strain, substrate, and growth phase (symbols, individual replicates). (C) Gene expression (as log₁₀ mean normalized counts) in *P. putida* ICE*clc* and derivative mutant strains in mixed carbon source, stationary phase condition, compared by computing pairwise correlations across all genes and fitting a linear model (dashed line). Colored enlarged dots indicate genes with significantly deviating expression among the paired comparison, calculated from the residuals of the linear model and multicomparison-corrected *t*-tests (Table S7). (D) Heatmap of expression of genes annotated with *cytochrome* in *P. putida* (as log₂ normalized counts).

This gene cluster contains two genes (*purU-III* and *folD-I*) that are predicted to participate in biosynthesis of mTHF, whereas the others have no detailed annotation, but may be responsible for carbon-hydroxylation and modification reactions based on low-confidence homology predictions (illustrated in Fig. 5C). The deletion of the gene fragment *purU-III* to PP_1946 proved lethal for *P. putida* UWC1, and a stable mutant could not be obtained (Fig. 5C). In contrast, two other deletions (i.e., PP_1947-PP_1951 and PP_1952-PP_1957) were obtained and could be tested for their effects on growth and ICE*clc* induction, alongside a mutant in which a second copy of the *purU-III* to PP_1946 genes was introduced via a mini-Tn*7* transposon. All mutants in this alternate mTHF pathway were still able to grow with 3-CBA as sole carbon and energy source although the PP_1947-PP_1951 deletion exhibited a longer lag phase and a lower maximum specific growth rate (Fig. S3). Importantly, every mutant showed lower proportions of tc cells in stationary phase conditions (Fig. 5D) and decreased ICE*clc* transfer rates compared to wild type (Fig. 5E, although the statistical significance of this decrease is less strong). These results suggest that ICE*clc* either senses metabolites produced by the alternate mTHF pathway or interacts with another intracellular signal that is modulated by this pathway.

## Different parts of ICE*clc* gene activation are affected in 3-CBA metabolic mutants

Finally, we noticed a number of curious expression differences in ICE*clc* core genes themselves among the mutants in the 3-CBA pathway, which may point to unresolved aspects of ICE activation and regulation. Consistent with its low proportion of tc cells (Fig. 3F) and reduced ICE-transfer rates (Fig. 3E), the Δ*clc* mutant displayed very poor expression of ICE core genes, whereas the Δ*ben* mutant showed slightly higher expression under stationary phase conditions with succinate + 3 CBA, compared to wild type (Fig. 6A, blue, salmon and green regions). In contrast, the complemented strain (Δ*clc*-Tn7*clc*) showed increased expression of the ICE core genes comparable to the wild type and in comparison to exponential and stationary phase on succinate (Fig. 6A, black symbols). In detail, however, genes in the regulatory module (*bisR-bisDC-ssb*) (19) were induced exclusively in *P. putida* ICE*clc* wild type, but not in the Δ*clc*- or Δ*ben*-mutants, and neither in the Δ*clc*-complemented strain (Fig. 6B, blue zone). In contrast, genes in the Type IV Secretion System module (*iceB7*, *iceD4,* and *pilT* chosen here as representatives) (29) were similarly induced in both wild type and the Δ*clc*-complemented strain (Fig. 6B, green-shaded zone). A set of further core genes of unknown function (*orf91884–orf74436*) showed higher expression in the Δ*clc*-complemented strain than in wild type and were even slightly induced in the Δ*ben*- compared to the Δ*clc*-mutant (Fig. 6B, salmon-shaded zone). This confirmed that, as expected, the ICE is not (or only slightly) activated in the absence of 3-CBA metabolism. Moreover, efficient ICE transfer may need strong expression of the regulatory modules (as in the wild-type, blue-shaded zone in Fig. 6).

## DISCUSSION

Conjugative transfer of ICE*clc* remains enigmatic in terms of the conditions that lead to its activation in a relatively small proportion of cells in stationary phase, particularly after growth on 3-CBA (Fig. 3F). Our combined mutant analysis and transcriptomic data demonstrate unequivocally that 3-CBA metabolism is required for ICE activation. Whereas prototrophic strains for 3-CBA metabolism activate ICE transfer after growth on 3-CBA (Fig. 3D), strains lacking either the initial conversion of 3-CBA to chlorocatechols (Δ*ben*-mutant, Fig. 1A) or the downstream chlorocatechol catabolism (Δ*clc*-mutant) fail to activate the ICE*clc* regulatory and type IV conjugation system genes (Fig. 6B), show significantly decreased proportions of tc cells (Fig. 3F), and display reduced transfer rates (Fig. 3E). Complementation of the Δ*clc*-deletion with a single copy insertion of the *clcRABDE*-genes at an independent locus on the *P. putida* chromosome restored growth on 3-CBA (Fig. 3A) but delayed the diauxic shift on succinate + 3 CBA and only partially

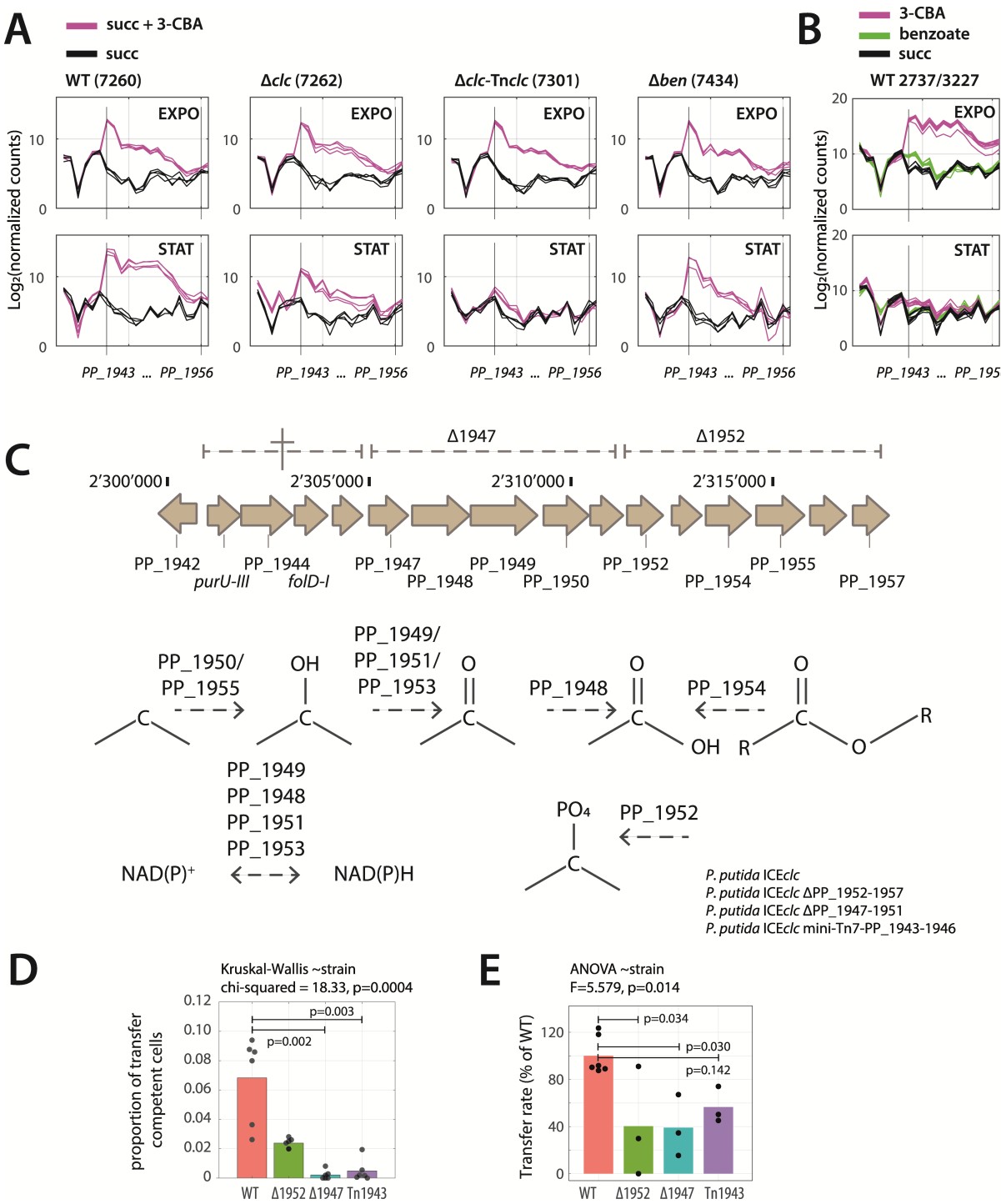

FIG 5 Involvement of an alternate pathway for methyltetrahydrofolate synthesis on ICE activation on 3-CBA. (A and B) Expression levels (as log$_2$ normalized counts) of the PP_1943-PP_1957 gene cluster in *P. putida* ICE*clc* wild type and derivative mutant strains across growth phase and substrates. (C) Genetic organization of PP_1943-PP_1957 gene cluster in *P. putida*. The deletions produced in this study are indicated with dashed lines. Proposed model of the reactions performed by the enzymes encoded in the PP_1943-PP_1957 gene cluster based on protein structure prediction and comparison. (D) Proportion of ICE*clc* transfer competent cells in *P. putida* wild-type ICE*clc* (2737), ΔPP_1952-PP_1957 deletion, ΔPP_1947-PP_1851 deletion, and Tn7-PP_1943-PP_1946, carrying the tc-fluorescent reporter P$_{inR}$-echerry grown 3 mM 3-CBA to late stationary phase. (E) ICE transfer frequency in proportion to the *P. putida* wild-type ICE*clc* (2737), for the ΔPP_1952-PP_1957 deletion, ΔPP_1947-PP_1851 deletion, and Tn7-PP_1943-PP_1946, grown on 3 mM 3-CBA.

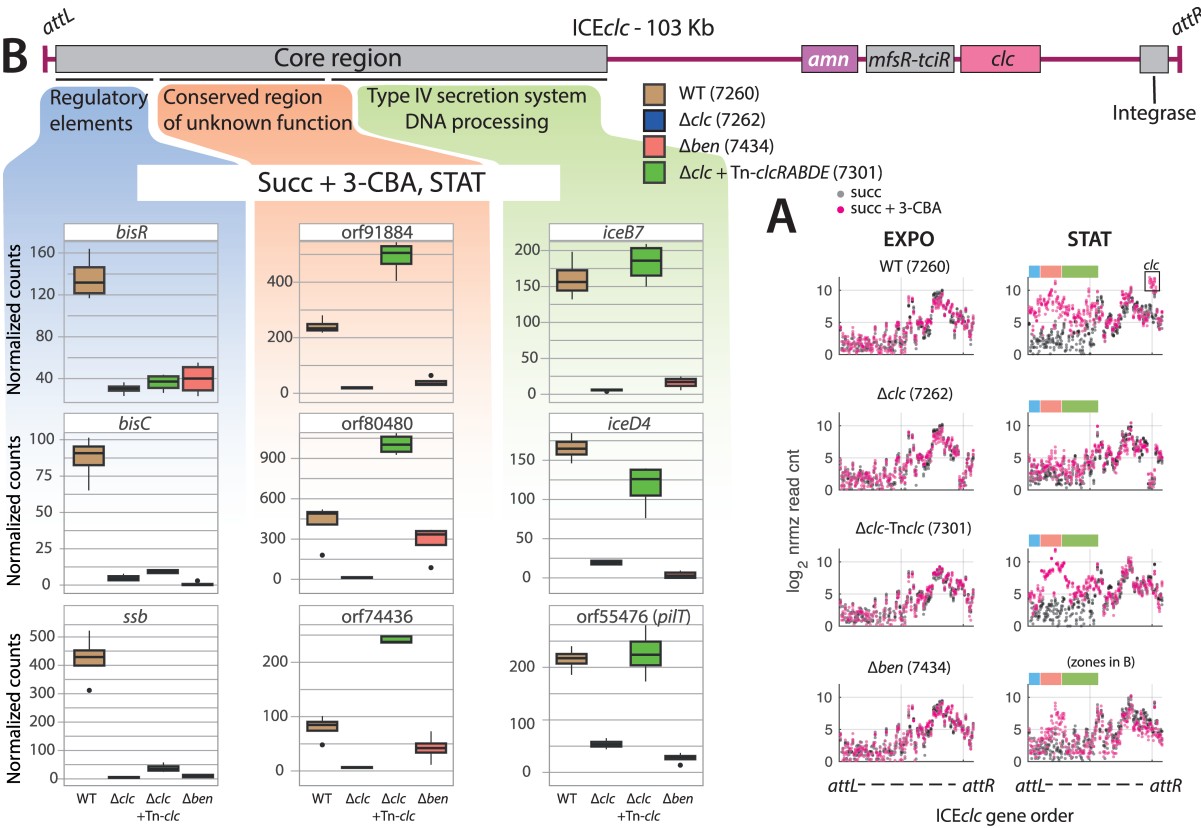

**FIG 6** ICEclc gene expression in 3-CBA metabolic pathway mutants. (A) Gene expression levels (as log2 normalized read counts) across the length of ICEclc in *P. putida* ICEclc and derivative mutant strains, in succinate or mixed carbon source, and exponential or stationary phase conditions. The ICEclc core region is indicated by color-coding corresponding to panel B. (B) Genetic organization of ICEclc core (gray) and metabolic (*amn* and *clc*) modules. The core region can be divided into "regulatory elements" (blue), "conserved region of unknown function" (salmon), and "Type IV Secretion System" (green). For each of these modules, expression levels are shown for three representative genes in *P. putida* ICEclc and derivative mutant strains, in mixed carbon source (succinate and 3-CBA) and stationary phase conditions.

induced ICEclc modules. Type IV conjugation system genes were expressed, whereas the regulatory cluster (*bisR*, *bisDC,* and *inrR-ssb*) remained silent (Fig. 6B). This suggests a previously unnoticed difference in the steps of ICE-activation, that we earlier considered to be comprised of 8–9 separately regulated gene clusters (24), to be activated in a distance-dependent manner from the origin of transfer of the ICE (30).

Despite 3-CBA metabolism being necessary for ICE-activation, 3-CBA or its known metabolites are unlikely to serve as the direct ICEclc transcriptional signal. Growth on succinate + 3 CBA failed to trigger ICE activation in Δclc- or Δben-mutant, and partial complementation with *clcR*, *clcRA,* or *clcRAB* did not raise ICE-transfer rates (Fig. 3E). We, therefore, assume that 3-CBA metabolism creates a physiological state indirectly sensed by ICE regulators such as TciR or BisR (19). To identify the characteristics of such a state, we compared global transcriptomes of wild-type and 3-CBA pathway mutants across multiple substrates and growth conditions. In contrast to 3-CBA, growth on its analog benzoate does not activate the ICE transfer competent state (Fig. S2). Clearly, growth on 3-CBA forces cells to operate differently than with its analog benzoate although there is crossregulation between both pathways which is independent of other genes on the ICE itself (Fig. 2). Transcriptomic data are suggestive of slower growth and different usage of respiratory pathways on 3-CBA as substrate. For example, levels of ribosomal protein and RNA polymerase transcripts remained higher for stationary phase conditions with 3-CBA than benzoate or succinate (Fig. 2C), and global metabolic pathway expression differed between benzoate and 3-CBA growth (Fig. 2A). Respiration with 3-CBA in exponential

phase caused different cytochromes to be prevalently expressed, some of which were specific with 3-CBA (e.g., *ccoNII-PII*), some others more reminiscent of stationary phase behavior with benzoate or succinate (Fig. 2E). In the absence of complete 3-CBA metabolism, these cytochrome expression signatures mostly disappeared (Fig. 4D). Both wild-type and *clc* deletion mutant showed higher summed transcript levels for oxidative stress in stationary phase when grown with 3-CBA, suggesting that oxidative stress signals may feed into ICE regulation (Fig. 4A and B).

Unexpectedly, 3-CBA strongly induces a polycistronic cluster bearing two genes implicated in mTHF biosynthesis (*purU-III* and *folD-I*, PP_1943-PP_1957). *P. putida* encodes three *purU*-genes, annotated as formyltetrahydrofolate deformylases (PP_0327, PP_1367, and PP_1943), and two *folD*-genes (5,10-methylene-tetrahydrofolate dehydrogenase, PP_1945 and PP_2265). Deletion attempts across *purU-III* and *folD-I* were lethal (Fig. 5B), confirming the essentiality of this sub-region, while the rest of the cluster tolerated removal. mTHF pathway induction is strong for growth on and/or exposure to 3-CBA while growing on succinate (Fig. 5A) but decreases in (real) stationary phase (Fig. 5B). Since we do not know what this pathway is specifically doing – it might be an alternate for mTHF biosynthesis or causing unknown other modifications to mTHF (Fig. 5C), we can only speculate what its role during growth on 3-CBA might be. Intriguingly, the mTHF cluster is induced even in the Δ*ben*-mutant that cannot transform 3-CBA, suggesting a defensive or repair response to 3-CBA-induced stress. Curiously, however, deletion mutants in this pathway (i.e., Δ1952 and Δ1947; Fig. 5C) showed lower proportions of ICE tc cells (Fig. 5D) and lower ICE transfer (Fig. 5E) despite normal growth on 3-CBA. Inserting a second copy of the first part of this pathway on a mini-transposon insertion into the *glmS* site (PP_1943-PP_1946) also lowered tc-cell percentages. This suggests that it might be a side effect of the mTHF pathway such as, for example, the maintenance of balance in oxidized vs reduced cofactors (NADH, NADPH) or the production of C1-metabolites, which influences ICE-activation.

Although our results show that growth on 3-CBA leads to a cellular state in stationary phase that is more prone to the activation of ICE*clc* transfer competence, there may be other conditions that provoke the same state. Previous studies have indicated that neither growth on other aromatic substrates (e.g., 4-hydroxybenzoate, benzoate, or anthranilate [22]) nor exposure to non-permissive temperatures, salt stress, ethanol stress, or ultraviolet light measurably induced ICE excision or ICE core promoter activation (31). However, we cannot exclude that other types of compounds leading to similar oxidative stress signals or mTHF induction would also make cells more prone to stationary phase ICE*clc* activation.

A hallmark of the ICE*clc* system is its bistable activation, occurring in only a sub-population during stationary phase (Fig. 5D). Transfer-competent cells appearing in stationary phase do not yet excise or transfer the ICE, which requires them to restart cell division (20, 21). Therefore, the specific ICE-gene expression we measure here in stationary phase cells (e.g., Fig. S2) is reflecting their expression from the integrated state in the subpopulation of cells and not from excised ICE-DNA molecules. Bistable activation is typically the sign of biochemical noise being amplified by the gene regulatory architecture, which could reflect some of the ICE key regulators for activation impinging on fluctuating cellular components. Earlier work demonstrated that this bistability in ICE*clc* relies on a cascade: MfsR represses *tciR* (32, 33); TciR activates *bisR*; BisR induces *bisDC*; BisDC then drives the expression of the downstream (or "late") ICE-clusters (19, 24, 34). BisD is a ParB-analog which clamps and slides on ICE-DNA, co-activating transcription and potentially facilitating ICE mobility (30). BisDC also autoregulates itself, which we hypothesized would enable bistability to persist for prolonged periods in stationary phase (19). Our data suggest an additional regulatory layer because *P. putida* with a deletion of the *clc*-genes on the ICE but complemented with a single copy mini-Tn7 insertion of the same *clc*-genes in the *glmS*-site (strain 7301) already induced the ICE-genes for the conjugative pores (e.g., *iceB7*, *iceD4*, *pilT*) and another part of the

ICE-core region (e.g., orf91884, orf80480) but fails to induce the *bisR, bisC,* and *ssb* module that are activated in the wild type (Fig. 6B).

This regulatory cascade, however, does not explain how environmental or intercellular cues, like 3-CBA-dependent metabolism, biases a subset of cells toward the tc-state. Our results here suggest that a physiological trigger such as elevated oxidative damage, altered respiration, or limited repair capacity is involved. This might lead to a common intracellular signal (e.g., redox state) that is perceived by TciR and helps to activate *bisR* that then elicits the bistable feedback of the ICE-activation cascade. Even such activation may not be direct but involve the balance in a number of additional factors, such as the previously reported importance in the levels of RpoS (35), which make it more likely for ICE-activation to occur. The difficulty here is really to achieve measurements of events at single cell resolution while preserving cell lineage history that would be necessary to understand the processes that control or favor horizontal gene transfer from elements such as ICE*clc*.

## MATERIALS AND METHODS

### Strains and culture conditions

*Escherichia coli* DH5α–λpir cells used for plasmid cloning were cultured at 37°C and 180 rpm rotary shaking in Luria-Bertani (LB) medium complemented with the appropriate antibiotic. The following antibiotic concentrations were used for the selection of genetic constructs: gentamicin (Gm) at 10 µg mL$^{-1}$, tetracycline (Tc) at 20 µg mL$^{-1}$, kanamycin (Km) at 25 µg mL$^{-1}$, and ampicillin (Amp) at 100 µg mL$^{-1}$. *Pseudomonas putida* UWC1-derived strains were grown at 30°C in LB with shaking at 180 rpm. Antibiotics included here were Gm at 20 µg mL$^{-1}$, Tc at 100 µg mL$^{-1}$, Km at 25–50 µg mL$^{-1}$, and Amp at 500 µg mL$^{-1}$. P$_{lac}$ induction of gene expression was achieved by adding 1 mM isopropyl β-D-1-thiogalactopyranoside (IPTG). Overnight *P. putida* cultures were 200-times diluted into 21°C minimal media (MM) (36) supplemented with 10 mM succinate, and re-incubated overnight at 30 °C with shaking at 180 rpm. The following day, cultures were again 200-times diluted into MM containing one of the following carbon sources: 10 mM succinate; 1 mM or 3 mM 3-CBA; 10 mM succinate + 3 mM 3-CBA; or 1 mM benzoate.

### Construction of *P. putida* strains

Seamless in-frame deletions of ICE*clc* or chromosomal genes were created using double recombination with marker counterselection, as described by Martìnez-Garcìa and de Lorenzo (37, 38). A ca. 1-kb section up- and downstream of the target loci was amplified from the *P. putida* genomic DNA using Q5 high fidelity polymerase (New England BioLabs) and primers as listed in Table S8. Both amplified fragments were then inserted in the linearized pEMG suicide vector (37, 38), by using the ClonExpress II one Step Cloning Kit (Vazyme). The plasmid was propagated in *E. coli* DH5a-λpir and purified (Macherey-Nagel NucleoSpin Plasmid kit), after which the insert was verified for correctness by sequencing. Plasmid DNA was then electroporated into *P. putida* UWC1 ICE*clc* strains. Transformants were selected on Km-resistance and examined by PCR to verify vector integration (Promega GoTaq Green Master Mix #M712). Single recombinants were regrown and transformed with plasmid pSW by electroporation (37, 38). After verification of the presence of pSW, transformants were induced with 15 mM *m*-toluate overnight to express SceI, leading to cleavage of inserted-pEMG and provoking the second recombination step. Culture dilutions were plated, and colonies were selected for the absence of Km-resistance and then verified by PCR for the deletion genotype. Strains with a successful deletion were passaged several times in the absence of antibiotic selection for loss of pSW and then stored at –80°C with 15% (vol/vol) glycerol.

ICE*clc* was tagged with a selection marker for transfer with a similar approach. The pEMG-Km$^R$ plasmid was inserted into the *amnB* gene of ICE*clc* without further recombination steps. After a first round of selection on Km, strains were passaged

twice successively in LB without antibiotics for loss of all non-integrated plasmids. The orientation of the insertion was verified by PCR.

Mini-Tn7 or Tn5 delivery was used for single-copy chromosomal complementation of deleted genes and for insertion of the $P_{inR}$–echerry tc-activation reporter (20). The loss of the pUX helper plasmid was confirmed by the absence of growth on 500 µg mL$^{-1}$ Amp. The correct insertion of the vector was verified by colony PCR, and the resulting strains were stored at −80 °C with 15% (vol/vol) glycerol.

In order to obtain partial complementation of the clc operon, the mini-Tn7 plasmid with the full clcRABDE genes described in reference 23 was digested to remove various parts of the clc genes as in Fig. 1, religated, verified, and then transformed into the relevant P. putida strains.

## RNA purification

Cultures were sampled for RNA isolation to obtain approximately $2.5 \times 10^8$ cells by collecting cells from exponential and stationary phase by centrifugation for 8 min at 2,612 relative centrifugal force (rcf). The culture liquid was decanted, and the cell pellet was immediately resuspended into 600 µL of TRI phenol reagent (Sigma-Aldrich) and stored at −80°C. RNA was extracted following the protocol of the Direct-zol RNA MiniPrep (Zymo Research). DNA traces were digested by treatment with TURBO DNase (Invitrogen by Thermo Fisher Scientific). RNA (>17 nt) was further purified and concentrated using the RNA clean and concentrator-5 (Zymo Research). RNA concentration and purity were estimated by absorbance at 260 nm and the absorbance ratios at 260/280 nm and 260/230 nm, by using a NanoDrop spectrophotometer (Witec AG), by Qubit with High Sensitivity (HS) dsDNA and RNA assays (Thermo Fisher Scientific) and by visualization on 1% agarose gel. When residual DNA was detected, a second DNaseI treatment was performed. DNA contamination was finally estimated by 25-cycles amplification of a 1-kb section upstream of the rpoS gene.

## RiboFree library preparation

Two hundred to five hundred nanograms of total RNA was reverse-transcribed, depleted from ribosomal RNAs, tagged on both ends and indexed, and then amplified by PCR using a Zymo-Seq RiboFree Total RNA Library Kit (Zymo Research R3000). After primer indexing, the mixture was cleaned up using 0.8-times concentrated Select-a-Size MagBeads (Zymo Research). The amount of double-stranded DNA (dsDNA) was measured with the Qubit fluorometer using the dsDNA HS (High Sensitivity) Assay Kit. Libraries were pooled in equimolar amounts, and primer dimers were removed by Select-a-Size Mag Beads cleaning. The quality and quantity of reverse-transcribed RNA were measured using a Fragment Analyzer 5200 (Agilent) at the Lausanne Genomic Technologies Facility, University of Lausanne, Switzerland. The RNAseq libraries were sequenced on an Illumina HiSeq4000 SR 150 platform (Illumina, Inc., San Diego, USA) or an Element AVITI platform.

## Bioinformatic tools

High-quality reads were mapped to the P. putida KT2440 reference annotated genome extracted from the Pseudomonas Genome Database (39) to which the ICEclc sequence was added at the appropriate tRNA-Gly position. Sequencing reads were first quality-trimmed with trimmomatic v0.39 (40) and aligned to the reference genome using STAR v2.7.10b (41). Samtools (v1.15.1) was used to remove PCR duplicates (42), and gene count tables were generated with HTSeq (v0.11.2) (43). Normalization of raw counts was performed using a geometric-mean size-factor procedure: a pseudo-reference profile was generated as the geometric mean of expression for each gene across all samples. After excluding genes with zero pseudo-reference values, sample-wise ratios to this reference were computed, and size factors were taken as the column-wise medians of these ratios. Raw counts were divided by the estimated size factors to obtain normalized

expression values. DESeq2 (v1.42.1) (44) was employed for differential expression and principal components analysis. Global gene expression values from *P. putida* ICE*clc* and derivative mutant strains (Fig. 4C; Table S8) were compared by computing pairwise correlations across all genes and fitting a linear model. Residuals were calculated for each gene, and a *t*-test was used to identify genes with expression significantly deviating from the wild-type-based prediction (the linear model). Heatmaps were generated using *pheatmap* (45). Permanova tests were performed with the *adonis2* function from the *vegan* package (46). AlphaFold3 (47) was used to predict protein structures, which were subsequently compared to the PDB using DALI (48). Pathway maps (Fig. 2A) were generated using iPath3.0 (49). StringDB-KEGG enrichment analysis was performed on the online STRING resource (50).

## ICE*clc* transfer

ICE*clc* transfer frequencies were measured from filter matings as described previously (31). The recipient strain was a *P. putida* UWC1 carrying a Tc-resistance cassette or a *P. putida* UWCGC carrying a Gm-resistance (Table S1). If not noted differently, donor strains were grown in MM with 3 mM 3-CBA for 48 h (with antibiotics and IPTG where relevant for selection of the genetic construct), while recipients were grown in MM with 5–10 mM succinate and Gm or Tc for 24 h. All strains were prepared in triplicates. Recipients and donors were then mixed in a 2:1 ratio (OD/OD) to a final volume of 1 mL. The cell mixtures were centrifuged for 2 min at 2,612 rcf, after which the supernatant was discarded and the cell pellet was washed in MM. After another round, the cells were resuspended in 20 µL of MM. This concentrated cell suspension was deposited on a 0.2–µm, 25 mm ø cellulose acetate filter (Huberlab, Aesch, Switzerland) placed on a MM-0.5 mM 3-CBA agar plate, or MM with 1 mM succinate (in case of Fig. 3D). After 48 h of incubation at 30℃, the cells were washed from filters with MM, diluted, and plated. Donors were selected and counted on MM agar plates with 3–5 mM 3-CBA, recipients on MM with 5–10 mM succinate + Gm or Tc. Transconjugants were selected on MM agar plates with 3 mM 3-CBA and Gm or Tc, or succinate, Km and Tc in case of transfers with the tagged ICE*clc* containing deletions in the *clc* genes. The ICE transfer frequency was then calculated as the number of transconjugants divided by the number of donors per milliliter volume in the same culture. Transfer rates were normalized across different experiments to that of the wild-type donor strain.

## Epifluorescence microscopy

Strains carrying the P$_{inR}$–*echerry* tc-activation reporter were grown in MM supplemented with 3 mM 3-CBA or 10 mM succinate + 3 mM 3-CBA, in biological triplicates, and harvested in late stationary phase. Images were taken on a Nikon Eclipse Ti-E inverted microscope, equipped with a perfect focus system, pE-100 CoolLED, and a Plan Apo λ 100× 1.45 oil objective (Nikon). The microscope was controlled using Micro-Manager (version 1.4) (http://www.micro-manager.org/) and snapshots were taken in Phase Contrast mode (50 ms exposure) and eCherry for P$_{inR}$–*echerry* reporter (500 ms). Images were segmented using Dimalis (51). Proportions of tc cells were calculated by quantile-quantile plotting as described in reference 22.

## ACKNOWLEDGMENTS

This work was supported by the Swiss National Science Foundation grants 31003A_175638 and 310030_204897 to J.R.V.D.M. Funding for the open access charge was provided by the Swiss National Science Foundation.

Conceptualization: R.B., V.B., J.R.V.D.M. Methodology: R.B., J.R.V.D.M. Investigation: R.B., V.B., H.B., A.C., V.S., A.D., N.C. Data curation: R.B., V.B. Formal analysis: R.B., V.B. Software: R.B., V.B. Validation: R.B., V.B., J.R.V.D.M. Visualization: R.B., V.B., J.R.V.D.M. Supervision: J.R.V.D.M. Project administration: J.R.V.D.M. Funding acquisition: J.R.V.D.M. Writing—

original draft: R.B., V.B., J.R.V.D.M. Writing—review and editing: R.B., V.B., H.B., A.C., V.S., A.D., N.C., J.R.V.D.M.

## AUTHOR AFFILIATION

[1]Department of Fundamental Microbiology, Université de Lausanne, Lausanne, Switzerland

## AUTHOR ORCIDs

Valentina Benigno ⓘ https://orcid.org/0009-0007-9493-0805
Nicolas Carraro ⓘ https://orcid.org/0000-0001-6364-547X
Jan Roelof van der Meer ⓘ http://orcid.org/0000-0003-1485-3082

## FUNDING

| Funder | Grant(s) | Author(s) |
| --- | --- | --- |
| Swiss National Science Foundation | 31003A_175638 | Jan Roelof van der Meer |
| Swiss National Science Foundation | 310030_204897 | Jan Roelof van der Meer |

## AUTHOR CONTRIBUTIONS

Roxane Bertholet, Conceptualization, Data curation, Formal analysis, Investigation, Methodology, Validation, Visualization, Writing – original draft, Writing – review and editing | Valentina Benigno, Conceptualization, Data curation, Formal analysis, Investigation, Methodology, Resources, Software, Validation, Visualization, Writing – original draft, Writing – review and editing | Hanna Budny, Investigation, Methodology | Anthony Convers, Investigation, Methodology | Vladimir Sentchilo, Investigation, Methodology, Supervision | Andrea Daveri, Investigation, Methodology, Writing – review and editing | Nicolas Carraro, Conceptualization, Formal analysis, Investigation, Methodology, Writing – original draft | Jan Roelof van der Meer, Conceptualization, Data curation, Formal analysis, Funding acquisition, Investigation, Methodology, Project administration, Resources, Software, Supervision, Validation, Visualization, Writing – original draft, Writing – review and editing

## DATA AVAILABILITY

The transcriptomics data sets generated and analyzed during the current study are available in the National Center for Biotechnology Information (NCBI) repository under BioProject accession number PRJNA784540 and the European Nucleotide Archive project number PRJEB106244.

## ADDITIONAL FILES

The following material is available online.

### Supplemental Material

**Supplemental material (mSystems00024-26-s0001.pdf).** Fig. S1-S3; Tables S1-S8.

### Open Peer Review

**PEER REVIEW HISTORY (review-history.pdf).** An accounting of the reviewer comments and feedback.

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
