## [Reviewer comments · mSystems]

Global transcriptomic profiling and mutant analysis suggest linked 3-chlorobenzoate metabolism and activation of an integrative and conjugative element in *Pseudomonas putida*

Roxane Bertholet, Valentina Benigno, Hanna Budny, Anthony Convers, Vladimir Sentchilo, Andrea Daveri, Nicolas Carraro, and Jan Roelof van der Meer

Corresponding Author(s): Jan Roelof van der Meer, Universite de Lausanne

Review Timeline:

Submission Date:	January 7, 2026
Editorial Decision:	January 27, 2026
Revision Received:	February 6, 2026
Accepted:	February 13, 2026

Editor: Jyot Antani

Reviewer(s): Disclosure of reviewer identity is with reference to reviewer comments included in decision letter(s). The following individuals involved in review of your submission have agreed to reveal their identity: Joao Botelho (Reviewer #2); Nathan Fraikin (Reviewer #3)

Transaction Report:

DOI: <https://doi.org/10.1128/msystems.00024-26>

Re: mSystems00024-26 (Global transcriptomic profiling and mutant analysis suggest linked 3-chlorobenzoate metabolism and activation of an integrative and conjugative element in *Pseudomonas putida*)

Dear Prof. Jan Roelof van der Meer:

I commend the authors on a well-written manuscript describing important findings. The reviewers appreciated the experimental approach and results, and I agree with this assessment. They raised specific comments that, once addressed, should further clarify the manuscript's narrative. Please incorporate the reviewer suggestions into your interpretation of the results and discussion.

Revision Guidelines

Sincerely,
Jyot Antani, PhD

Reviewer #1 (Comments for the Author):

In this study, the authors sought to elucidate the mechanism underlying 3-CBA induced ICE_{clc} transfer, mainly through the use of 3-CBA pathway mutants' analysis and RNA-seq analyses. Conjugation assays and transfer-competent cell counts using these mutants support 3-CBA metabolism as the main trigger for ICE_{clc} transfer. RNA-seq analysis further indicated that the wild-type strain displays substantially higher expression of Type IV secretion system genes involved in conjugation than the mutant strains

in the presence of 3-CBA. The authors further identified a novel link between mTHF pathway expression and ICEclc transfer induction. In addition, the RNA-seq analysis is comprehensive, spanning different carbon sources, strains, and growth phases, and reveals several interesting findings not directly related to conjugation activation, such as the upregulation of cytochrome gene clusters in benzoate- vs 3-CBA-grown cells and crosswise pathway interference between benzoate and 3-CBA degradation pathways. Overall, the study elucidates ICEclc transfer induction by 3-CBA and the host bacterial response to this compound. Despite the large amount of data presented, the Results section is concise and well organized, and the clear figures greatly facilitate understanding. In my view, this is a very thoughtful and well-executed study. I found only a few minor points that should be addressed or considered by the authors.

Major point

- The manuscript does not address key aspects of the ICE life cycle, namely excision from the host chromosome and, in some cases, subsequent replication. The transcriptional activation of conjugation-related genes observed in response to 3-CBA metabolism could potentially be explained by increased ICE excision and/or copy number. It may therefore be worthwhile for the authors to assess, or at least discuss, the excision frequency and copy number of the ICE under different carbon source conditions and in the mutant strains used in this study, in order to clarify whether changes in ICE abundance contribute to the observed transcriptional effects.
- It is unclear whether DNA transfer induction depends on the growth phase (stationary versus exponential). When cells were grown on 3-CBA + succinate, increased transcription of T4SS-related genes was observed only in the stationary phase (Fig. 6A), which may simply reflect that 3-CBA is metabolized mainly during this phase. Indeed, the authors assume that 3-CBA metabolism occurs around the stationary phase. It would therefore be informative to examine whether growth on 3-CBA as the sole carbon source also leads to increased expression of T4SS-related genes during the exponential phase. This question could potentially be addressed using the RNA-seq datasets already generated in this study.

Minor point

Fig. 1A "X-some" is not a standard term in genetics, and replacing it with "chromosome" would improve clarity and consistency.

Fig. 3A The labeling of the references/conditions would benefit from clarification, as "WT" appears to be used for two different cases.

P19 L21 - 22 Mating assays were performed on CBA agar plates. However, in Fig. 3D, donor cells were grown on succinate. It is therefore unclear whether CBA agar plates were also used for the mating experiments in this condition, or whether mating was performed on succinate agar plates instead. Please clarify this point.

P16 L16 For clarity, "LB" should be defined as Luria-Bertani medium at its first occurrence (p.16, line 12).

Reviewer #2 (Comments for the Author):

The paper is well-written and appropriately concise. The methods are clearly described, and the results provide compelling evidence for the link between 3-CBA metabolism and ICEclc activation in *Pseudomonas putida*. However, there are some limitations that should be addressed:

- The paper demonstrates that 3-CBA metabolism is necessary for ICE activation. While all experiments test necessity within the 3-CBA metabolic context, none test whether alternative physiological perturbations, such as imposed redox stress, treatment with other xenobiotics or manipulation of respiratory pathways independent of 3-CBA, can similarly activate ICE transfer. Without sufficiency tests conducted across different metabolic contexts, the conclusion risks generalizing a 3-CBA-specific phenomenon to a general principle of metabolic state-dependent ICE activation. The authors should acknowledge this limitation and clarify that, while their results identify specific physiological signatures associated with 3-CBA metabolism, such as altered respiration, oxidative stress and mTHF pathway induction, it remains to be determined whether these or other metabolic states can independently trigger ICE activation.

- The paper does not quantify 3-CBA consumption or chlorocatechol production directly at the time points used for ICE activation assays and transcriptome sampling. Without metabolite or flux measurements, it remains indirect to infer that differences in ICE activation are due to active 3-CBA catabolism rather than mere exposure-induced stress responses. This is particularly relevant given the substantial timing differences among strains during diauxic growth, which could confound the comparisons. Conducting targeted metabolomics or isotope tracing to confirm that 3-CBA conversion rates and chlorocatechol levels coincide with the observed differences in ICE activation would strengthen the causal link between metabolism (rather than exposure) and ICE induction. The authors should acknowledge that, without such measurements, the mechanism linking catabolism to ICE activation remains correlative.

Reviewer #3 (Comments for the Author):

The ms by Bertholet, Benigno et al investigates the effect of 3-chlorobenzoate metabolism on the activation of the ICEclc mobile

genetic element, which itself encodes enzymes that funnel this compound into the catechol degradation degradation pathway. The ms offers an interesting insight into the activation of this ICE by metabolic cues, especially due to the environmental and bioremediation implications of these compounds. However, the main message and questions of this manuscript are obtuse and difficult to disentangle from the massive amount of transcriptomic data provided in the manuscript. In my opinion, the ms would highly benefit from being streamlined and simplified to facilitate comprehension. For example, the point and the questions that would be answered from a transcriptomic analysis on all three substrates (Fig 1) are not clearly in the ms. Similarly, the point of running a deep metabolic analysis of the transcriptome (Fig 2a) or a GO analysis (Fig 2c) is highly convoluted and I was not able to discern evident and relevant conclusions for these data. I would therefore encourage the authors to revise the structure of the ms to a more streamlined structure. For example, the ms could start by presenting the DE between BEN and 3-CBA conditions (Fig 2b), tunnelling the rest of the ms to study genes identified in this analysis (eg cytochromes, noncanonical thf pathway etc) and their effects on ICE activation and transfer rate, while relegating the overwhelming amount of transcriptomic analysis to supplementary data.

Minor comments :

Fig 1 : Benzoate activates the clc pathway but reasons behind this activation are not discussed and evaluated further. For example, would benzoate also activate ICEclc transfer ?

Fig 2c : induction of alternative cytochromes and of the mTHF island (likely implicated in 1-carbon metabolism) would indicate that a product of 3-CBA metabolism is metabolized through an alternative, detoxifying pathway ?

Fig 5e : The effect of mTHF pathway gene deletions on plasmid transfer seem rather unconvincing given that most other effects in the ms are reported on a log scale (eg Fig 3d-e) while these are reported on a linear scale with a weak statistical significance and the use of a parametric test for data that does not look robustly distributed. Could the authors check the normality and homoscedasticity of the data and apply relevant tests (Welch ANOVA or Kruskal-Wallis).

In this study, the authors sought to elucidate the mechanism underlying 3-CBA induced ICE*clc* transfer, mainly through the use of 3-CBA pathway mutants' analysis and RNA-seq analyses. Conjugation assays and transfer-competent cell counts using these mutants support 3-CBA metabolism as the main trigger for ICE*clc* transfer. RNA-seq analysis further indicated that the wild-type strain displays substantially higher expression of Type IV secretion system genes involved in conjugation than the mutant strains in the presence of 3-CBA. The authors further identified a novel link between mTHF pathway expression and ICE*clc* transfer induction. In addition, the RNA-seq analysis is comprehensive, spanning different carbon sources, strains, and growth phases, and reveals several interesting findings not directly related to conjugation activation, such as the upregulation of cytochrome gene clusters in benzoate- vs 3-CBA-grown cells and crosswise pathway interference between benzoate and 3-CBA degradation pathways. Overall, the study elucidates ICE*clc* transfer induction by 3-CBA and the host bacterial response to this compound. Despite the large amount of data presented, the Results section is concise and well organized, and the clear figures greatly facilitate understanding. In my view, this is a very thoughtful and well-executed study. I found only a few minor points that should be addressed or considered by the authors.

Major point

- The manuscript does not address key aspects of the ICE life cycle, namely excision from the host chromosome and, in some cases, subsequent replication. The transcriptional activation of conjugation-related genes observed in response to 3-CBA metabolism could potentially be explained by increased ICE excision and/or copy number. It may therefore be worthwhile for the authors to assess, or at least discuss, the excision frequency and copy number of the ICE under different carbon source conditions and in the mutant strains used in this study, in order to clarify whether changes in ICE abundance contribute to the observed transcriptional effects.
- It is unclear whether DNA transfer induction depends on the growth phase (stationary versus exponential). When cells were grown on 3-CBA + succinate, increased transcription of T4SS-related genes was observed only in the stationary phase (Fig. 6A), which may simply reflect that 3-CBA is metabolized mainly during

this phase. Indeed, the authors assume that 3-CBA metabolism occurs around the stationary phase. It would therefore be informative to examine whether growth on 3-CBA as the sole carbon source also leads to increased expression of T4SS-related genes during the exponential phase. This question could potentially be addressed using the RNA-seq datasets already generated in this study.

Minor point

Fig. 1A “X-some” is not a standard term and replacing it with “chromosome” would improve clarity and consistency.

Fig. 3A The labeling of the references/conditions should be clarification, as “WT” appears to be used for two different cases.

P19 L21 – 22 Mating assays were performed on CBA agar plates. However, in Fig. 3D, donor cells were grown on succinate. It is therefore unclear whether CBA agar plates were also used for the mating experiments in this condition, or whether mating was performed on succinate agar plates instead. Please clarify this point.

P16 L16 For clarity, “LB” should be defined as *Luria–Bertani medium* at its first occurrence (p.16, line 12).

Rebuttal letter

Re: mSystems00024-26 (Global transcriptomic profiling and mutant analysis suggest linked 3-chlorobenzoate metabolism and activation of an integrative and conjugative element in *Pseudomonas putida*)

Line and page numbering correspond to the PDF with all changes marked.

Reviewer #1 (Comments for the Author):

In this study, the authors sought to elucidate the mechanism underlying 3-CBA induced ICEclc transfer, mainly through the use of 3-CBA pathway mutants' analysis and RNA-seq analyses. Conjugation assays and transfer-competent cell counts using these mutants support 3-CBA metabolism as the main trigger for ICEclc transfer. RNA-seq analysis further indicated that the wild-type strain displays substantially higher expression of Type IV secretion system genes involved in conjugation than the mutant strains in the presence of 3-CBA. The authors further identified a novel link between mTHF pathway expression and ICEclc transfer induction. In addition, the RNA-seq analysis is comprehensive, spanning different carbon sources, strains, and growth phases, and reveals several interesting findings not directly related to conjugation activation, such as the upregulation of cytochrome gene clusters in benzoate- vs 3-CBA-grown cells and crosswise pathway interference between benzoate and 3-CBA degradation pathways. Overall, the study elucidates ICEclc transfer induction by 3-CBA and the host bacterial response to this compound. Despite the large amount of data presented, the Results section is concise and well organized, and the clear figures greatly facilitate understanding. In my view, this is a very thoughtful and well-executed study. I found only a few minor points that should be addressed or considered by the authors.

We thank the reviewer for the summary of our work and for the overall positive impression of our study.

Major point

- The manuscript does not address key aspects of the ICE life cycle, namely excision from the host chromosome and, in some cases, subsequent replication. The transcriptional activation of conjugation-related genes observed in response to 3-CBA metabolism could potentially be explained by increased ICE excision and/or copy number. It may therefore be worthwhile for the authors to assess, or at least discuss, the excision frequency and copy number of the ICE under different carbon source

conditions and in the mutant strains used in this study, in order to clarify whether changes in ICE abundance contribute to the observed transcriptional effects.

*We thank the reviewer for the interesting question. The subpopulation-dependent formation of transfer-competent cells makes all analyses a bit more complicated, but here is what we know. Previous time-lapse microscopy imaging with single copy gene reporter constructs to ICE-promoters showed that activation of transfer competence in individual cells occurs in stationary phase cultures (only visible when cultures have been grown on 3CBA), but ICE excision only takes place **after** ‘activated’ cells acquire new nutrients and resume growth¹. We repeated measurements of single cell single copy reporters fused to 11 presumed promoters of the ICEclc core region alone and in combination, which indicated they all fire in the same cell². The ICE transfer-competence program is thus coherent for a subgroup of cells and single copy gene reporters to key ICE promoters (e.g., the integrase gene) are a good proxy to follow the activation process.*

Follow-up experiments using single cell measurements of individual cell ICE copy numbers by LacI-fluorescent protein fusions indeed demonstrated that there is also limited replication of excised ICE, but this is only taking place when pre-activated transfer-competent cells resume growth³. That study also indicated that there is no ICE excision in cells that have not pre-expressed the ICE transfer competent genes. More recently we showed that this also holds for the formation of the ICE conjugation complexes; they form exclusive in tc-cells and are produced before ICE excision or transfer⁴.

Indeed, as the reviewer suggested, single cell studies also indicated that there is a difference in the expression of (some) ICE genes in the excised form compared to the integrated form, most notably because excision and DNA-recircularization places a strong promoter in front of the integrase gene¹.

This all leads to the same conclusion: the observed expression of ICE-core genes in bulk population is representative for expression in a small proportion of tc cells (3-5% of the population), and expression of transfer competence genes precedes ICE excision.

¹ Delavat F, Mitri S, et al. (2016). Highly variable individual donor cell fates characterize robust horizontal gene transfer of an integrative and conjugative element. Proc. Natl. Acad. Sci. U. S. A. 113: E3375-3383. doi:10.1073/pnas.1604479113.

² Sulser S, Vucicevic A, et al. (2022). A bistable prokaryotic differentiation system underlying development of conjugative transfer competence. PLoS Genet 18: e1010286. doi:10.1371/journal.pgen.1010286.

³ Delavat F, Moritz R, et al. (2019). Transient replication in specialized cells favors transfer of an Integrative and Conjugative Element. mBio 10. doi:10.1128/mBio.01133-19.

⁴ Daveri A, Benigno V, et al. (2023). Characterization of an atypical but widespread type IV secretion system for transfer of the integrative and conjugative element (ICEclc) in Pseudomonas putida. Nucleic Acids Res 51: 2345-2362. doi:10.1093/nar/gkad024.

Therefore, under the experimental conditions deployed here (RNA isolation from stationary phase cells), there is no ICE excision yet and thus the differences in observed (bulk) expression levels are the result of differences in ICE gene activation and differences in the subpopulation size of cells that have initiated the tc program.

Action: we have added this to the introduction on p.5, l. 33 – p. 6 l. 3 and the discussion on p. 16, l. 2-8.

- It is unclear whether DNA transfer induction depends on the growth phase (stationary versus exponential). When cells were grown on 3-CBA + succinate, increased transcription of T4SS-related genes was observed only in the stationary phase (Fig. 6A), which may simply reflect that 3-CBA is metabolized mainly during this phase. Indeed, the authors assume that 3-CBA metabolism occurs around the stationary phase. It would therefore be informative to examine whether growth on 3-CBA as the sole carbon source also leads to increased expression of T4SS-related genes during the exponential phase. This question could potentially be addressed using the RNA-seq datasets already generated in this study.

We thank the reviewer for bringing up this point and apologize if this has not become clear. Appearance of tc cells under wild-type genotype exclusively and only occurs under stationary phase conditions¹ and is RpoS-dependent⁵. However, actual ICE transfer from tc cells requires reactivation of their cell cycle in presence of fresh nutrients³. In order to study the effect of 3CBA metabolism we had to make mutants that no longer grow on 3CBA, but these require incubation with both succinate (for growth) and 3CBA (for potential effects of intermediates). Indeed, 3CBA growth of the wild-type leads to formation of tc cells in stationary phase and only then is ICE core expression elevated². Fig 3D shows the effect of growth on 3CBA as opposed to succinate on ICE transfer in wild-type cells not affected in 3CBA metabolism.

Action: We have rephrased these points for clarity in the discussion on page 14 (l. 25 and 27/28).

Minor point

Fig. 1A "X-some" is not a standard term in genetics, and replacing it with "chromosome" would improve clarity and consistency.

We thank the reviewer. This was corrected in the revision of Fig. 1.

⁵ Miyazaki R, Minoia M, et al. (2012). Cellular variability of RpoS expression underlies subpopulation activation of an integrative and conjugative element. *PLoS Genet.* 8: e1002818. doi:10.1371/journal.pgen.1002818.

Fig. 3A The labeling of the references/conditions would benefit from clarification, as "WT" appears to be used for two different cases.

We thank the reviewer. We verified this, but have consistently used WT here to refer to the wild-type prototroph in terms of the 3CBA pathway (strain 7260). This was specified everywhere in the figure and the legend.

P19 L21 - 22 Mating assays were performed on CBA agar plates. However, in Fig. 3D, donor cells were grown on succinate. It is therefore unclear whether CBA agar plates were also used for the mating experiments in this condition, or whether mating was performed on succinate agar plates instead. Please clarify this point.

We thank the reviewer. We specified this further. In case of succinate-grown donor cells, the mating plates also contained succinate. Selection of transconjugants (in Fig. 3D) was done on the Km-antibiotic resistance marker of the ICE, as indicated by the y-axis label (Km-r-transconjugants per donor).

P16 L16 For clarity, "LB" should be defined as Luria-Bertani medium at its first occurrence (p.16, line 12).

We thank the reviewer. This was added.

Reviewer #2 (Comments for the Author):

The paper is well-written and appropriately concise. The methods are clearly described, and the results provide compelling evidence for the link between 3-CBA metabolism and ICE_{clc} activation in *Pseudomonas putida*. However, there are some limitations that should be addressed:

- The paper demonstrates that 3-CBA metabolism is necessary for ICE activation. While all experiments test necessity within the 3-CBA metabolic context, none test whether alternative physiological perturbations, such as imposed redox stress, treatment with other xenobiotics or manipulation of respiratory pathways independent of 3-CBA, can similarly activate ICE transfer. Without sufficiency tests conducted across different metabolic contexts, the conclusion risks generalizing a 3-CBA-specific phenomenon to a general principle of metabolic state-dependent ICE activation. The authors should acknowledge this limitation and clarify that, while their results identify specific physiological signatures associated with 3-CBA metabolism, such as altered

respiration, oxidative stress and mTHF pathway induction, it remains to be determined whether these or other metabolic states can independently trigger ICE activation.

We thank the reviewer for bringing up this point. Indeed, our focus here was specifically the question of how 3-CBA metabolism could be influencing the formation of tc cells and we did not specifically test other conditions that could potentially lead to tc cell activation (except benzoate).

*However, in previous studies we examined a variety of conditions and showed they have no effect on ICE activation, excision and transfer, among which: temperature shock (42°C; 37°C; 4°C), exposure to monochlorobenzene; 2 or 5% ethanol; UV exposure at 2, 10 or 50 J/cm²⁶; salt stress (0.2 and 0.5 M NaCl); benzoate, 4-OH-benzoate, or anthranilate⁷ (all growth substrates for *P. putida* and *P. knackmussii* strain B13 – the original donor of ICEclc).*

*In a related submission of our group⁸ we further show how neither antibiotics, heavy metals, or oxidative stressors activate ICE excision in *P. aeruginosa* and *P. putida*.*

Action: We specify these points and address the limitations in a new paragraph in the discussion on p. 15, l. 26-32.

- The paper does not quantify 3-CBA consumption or chlorocatechol production directly at the time points used for ICE activation assays and transcriptome sampling. Without metabolite or flux measurements, it remains indirect to infer that differences in ICE activation are due to active 3-CBA catabolism rather than mere exposure-induced stress responses. This is particularly relevant given the substantial timing differences among strains during diauxic growth, which could confound the comparisons. Conducting targeted metabolomics or isotope tracing to confirm that 3-CBA conversion rates and chlorocatechol levels coincide with the observed differences in ICE activation would strengthen the causal link between metabolism (rather than exposure) and ICE induction. The authors should acknowledge that, without such measurements, the mechanism linking catabolism to ICE activation remains correlative.

We thank the reviewer for (again) another very valid question. Indeed, we have not measured 3-CBA consumption or chlorocatechol production in these assays, which, unfortunately, are assays that our laboratory is not immediately equipped for. However,

⁶ Sentschilo VS, Ravatn R, et al. (2003). Unusual integrase gene expression on the clc genomic island of *Pseudomonas* sp. strain B13. *J. Bacteriol.* 185: 4530-4538. doi:

⁷ Reinhard F and van der Meer JR (2013). Improved statistical analysis of low abundance phenomena in bimodal bacterial populations. *PLoS ONE* 8: e78288. doi:10.1371/journal.pone.0078288.

⁸ Benigno V, Carraro N, et al. (2026). Horizontal transfer of ICEclc-like elements in *Pseudomonas aeruginosa* clinical isolates. *bioRxiv* 2026.01.08.698369

we respectfully disagree that active 3-CBA metabolism would not be necessary for ICE activation for the following reasons:

Mutants in either part of the 3-CBA pathway (benzoate dioxygenase or chlorocatechol 1,2-dioxygenase) do not induce the ICE. Therefore, simple exposure to 3-CBA (as in the Δben mutant) or to chlorocatechols (as in the Δclc mutant), or to further intermediates of chlorocatechols (as in the complemented Δclc mutants) do not trigger ICE activation.

Furthermore, growth on 3-CBA by the wild-type triggers ICE activation (but only in stationary phase), but growth on benzoate does not. We believe this justifies the conclusion the metabolism of (and not exposure to) 3-CBA is needed to produce the metabolic state that favors formation of tc cells.

Action: We rewrote this part of the discussion to articulate this more clearly p. 14, l. 25-28.

Reviewer #3 (Comments for the Author):

The ms by Bertholet, Benigno et al investigates the effect of 3-chlorobenzoate metabolism on the activation of the ICE_{clc} mobile genetic element, which itself encodes enzymes that funnel this compound into the catechol degradation pathway. The ms offers an interesting insight into the activation of this ICE by metabolic cues, especially due to the environmental and bioremediation implications of these compounds.

However, the main message and questions of this manuscript are obtuse and difficult to disentangle from the massive amount of transcriptomic data provided in the manuscript. In my opinion, the ms would highly benefit from being streamlined and simplified to facilitate comprehension.

We thank this reviewer for the comment and apologize if the data presentation at times was difficult to follow (although both other reviewers did not share this aspect). We already removed various aspects of the transcriptomic data presentation.

Action: no further action needed.

For example, the point and the questions that would be answered from a transcriptomic analysis on all three substrates (Fig 1) are not clearly in the ms. Similarly, the point of running a deep metabolic analysis of the transcriptome (Fig 2a) or a GO analysis (Fig 2c) is highly convoluted and I was not able to discern evident and relevant conclusions for these data.

*We thank the reviewer for bringing up this comment. We use the data of the wild-type strain because otherwise arguments like advanced by reviewer 2 would make sense. The use of double substrates (forced by the use of mutants in 3-CBA metabolism) is not ideal and leads to diauxic growth which may have other interpretations. For this reason, the wild-type data are necessary. The comparison of benzoate and 3-CBA is important because they are analogous substrates. It might also be that other genes on the ICE influence host expression and confound our interpretation of results. This is why we also added the analysis of *P. putida* without ICE but with the *clc* genes.*

Since we find that not a single set of genes is responsible for associating the 3-CBA effect to ICE-activation in stationary phase, we think it is important to try and describe the differences in global metabolic states of cells under the various conditions. This automatically leads to some convolution because of using tools such as GO or KEGG pathway analysis that deploy more general terms for gene expression differences.

Action: We rephrased on p. 15, l. 25-32

I would therefore encourage the authors to revise the structure of the ms to a more streamlined structure. For example, the ms could start by presenting the DE between BEN and 3-CBA conditions (Fig 2b), tunnelling the rest of the ms to study genes identified in this analysis (eg cytochromes, noncanonical thf pathway etc) and their effects on ICE activation and transfer rate, while relegating the overwhelming amount of transcriptomic analysis to supplementary data.

Action: We have added several parts in the discussion to clarify this (as shown in the annotated revised version).

Minor comments :

Fig 1 : Benzoate activates the *clc* pathway but reasons behind this activation are not discussed and evaluated further. For example, would benzoate also activate ICE_{clc} transfer ?

We thank the reviewer for the comment. We apologize if this was not clear. Benzoate does not activate ICE_{clc} transfer nor leads to the formation of tc cells.

Action: We added a new supplementary figure showing just the expression of the ICE genes in exponential and stationary phase in wild-type cells and all three substrate conditions (Fig. S2).

Fig 2c : induction of alternative cytochromes and of the mTHF island (likely implicated in 1-carbon metabolism) would indicate that a product of 3-CBA metabolism is metabolized through an alternative, detoxifying pathway ?

We thank the reviewer for this question. Unfortunately, we do not know this, but this is what we speculate in the discussion (p. 14, l.6-24)

Fig 5e : The effect of mTHF pathway gene deletions on plasmid transfer seem rather unconvincing given that most other effects in the ms are reported on a log scale (eg Fig 3d-e) while these are reported on a linear scale with a weak statistical significance and the use of a parametric test for data that does not look robustly distributed. Could the authors check the normality and homoscedasticity of the data and apply relevant tests (Welch ANOVA or Kruskal-Wallis).

We thank the reviewer for the comment – statistics being what it is. Quantification of the proportion of transfer competent cells in the mTHF mutants are statistically significantly lower than wild-type (Fig. 3D) and this is in agreement with the ICE transfer rates (Fig. 3E). The P-values of the data in Fig. 3E are higher than in Fig. 3D and there is more replicate variation for the transfer rates. The statistics of the data analysis is as implemented in R – anova. Shapiro test for the residuals was okay and Levene’s test for homogeneity of variances was also correct for running the ANOVA. The P-values come from post-Tukey HSD testing, as usual and we report the adjusted-P from that test.

Action: We added the (non-significant) P-value for the comparison to the Tn7-PP1943 mutant, and mentioned the lower confidence in the results section.

Re: mSystems00024-26R1 (Global transcriptomic profiling and mutant analysis suggest linked 3-chlorobenzoate metabolism and activation of an integrative and conjugative element in *Pseudomonas putida*)

Dear Prof. Jan Roelof van der Meer,

Congratulations! Your manuscript has been accepted, and I am forwarding it to the ASM production staff for publication. Your paper will first be checked to make sure all elements meet the technical requirements. ASM staff will contact you if anything needs to be revised before copyediting and production can begin. Otherwise, you will be notified when your proofs are ready to be viewed.

Sincerely,
Jyot Antani, PhD

Reviewer #1 (Comments for the Author):

The authors have thoughtfully addressed all of my previous concerns, and the revisions have significantly strengthened the paper. I have no further suggestions for revision.

Reviewer #2 (Comments for the Author):

The authors made all of the suggested revisions, and I have no further comments.